# Repetitive Anodal TDCS to the Frontal Cortex Increases the P300 during Working Memory Processing

**DOI:** 10.3390/brainsci12111545

**Published:** 2022-11-14

**Authors:** Angela Voegtle, Christoph Reichert, Hermann Hinrichs, Catherine M. Sweeney-Reed

**Affiliations:** 1Neurocybernetics and Rehabilitation, Department of Neurology, Otto von Guericke University, 39120 Magdeburg, Germany; 2Department of Behavioral Neurology, Leibniz Institute for Neurobiology, 39118 Magdeburg, Germany; 3Center for Behavioral Brain Sciences—CBBS, Otto von Guericke University, 39106 Magdeburg, Germany; 4Department of Neurology, Otto von Guericke University, 39120 Magdeburg, Germany

**Keywords:** TDCS, repetitive, working memory, n-back, ERP, P300

## Abstract

Transcranial direct current stimulation (TDCS) is a technique with which neuronal activity, and therefore potentially behavior, is modulated by applying weak electrical currents to the scalp. Application of TDCS to enhance working memory (WM) has shown promising but also contradictory results, and little emphasis has been placed on repeated stimulation protocols, in which effects are expected to be increased. We aimed to characterize potential behavioral and electrophysiological changes induced by TDCS during WM training and evaluate whether repetitive anodal TDCS has a greater modulatory impact on the processes underpinning WM than single-session stimulation. We examined the effects of single-session and repetitive anodal TDCS to the dorsolateral prefrontal cortex (DLPFC), targeting the frontal-parietal network, during a WM task in 20 healthy participants. TDCS had no significant impact on behavioral measures, including reaction time and accuracy. Analyzing the electrophysiological response, the P300 amplitude significantly increased following repetitive anodal TDCS, however, positively correlating with task performance. P300 changes were identified over the parietal cortex, which is known to engage with the frontal cortex during WM processing. These findings support the hypothesis that repetitive anodal TDCS modulates electrophysiological processes underlying WM.

## 1. Introduction

Transcranial direct current stimulation (TDCS) is a noninvasive technique, with which a weak direct current is applied noninvasively over the scalp to alter underlying cortical excitability. Anodal TDCS results in a change in the resting membrane potential towards depolarization, which increases the spontaneous firing of neurons [1,2,3]. It has gained increasing attention in recent years as a potential tool with which to modulate cognitive processing [4,5,6] and has been shown to modify behavioral performance in a range of cognitive tasks, including working memory (WM) processing. The term WM refers to a set of basic, rapidly accessible cognitive functions with limited capacity, which temporarily store and update information as well as manipulate it for use in higher cognitive processes, such as language comprehension, learning, and reasoning [7,8,9], which are crucial to daily life [9]. Although results have been mixed [10,11,12,13,14], a number of studies have been reported in which accuracy and/or reaction times (RTs) have indicated enhanced WM performance during or immediately after TDCS [11,12,13]. The effects are generally of short duration [2], but because the effect of TDCS can outlast the stimulation period itself [2,3,14,15,16], repetitive TDCS could have a cumulative effect, increasing its influence on cortical networks. Therefore, a growing body of research is investigating the influence of repetitive TDCS, with the findings so far suggesting a benefit in WM performance [17,18,19,20,21,22]. Contradictory and null results have also been reported, however [18,23,24,25]. Moreover, the neural correlates of the performance improvement were not explored in all the studies reporting positive effects.

The dorsolateral prefrontal cortex (DLPFC) has been found to be consistently active during WM paradigms [26] and shows increased activity during demanding WM tasks [27]. Its functionality appears to be lateralized, with the left DLPFC being relevant for verbal tasks [26,27,28]. Another key region in WM processing is the posterior parietal cortex (PPC) [20,26,27,29,30,31,32]. Evidence suggests that a frontal-parietal network involving DLPFC and PPC underlies WM function, integrating information from posterior regions, where mainly storage takes place, and anterior regions, congregating in the DLPFC, where executive processes are located [33]. In event-related potential (ERP) studies, the late positive component (P300) has been consistently associated with WM [34,35,36] in different tasks such as the oddball [36], n-back [37,38], and Go/NoGo tasks [39]. It has multiple cortical and subcortical generators [40], including the DLPFC [41,42], with a posterior component (P3b) reflecting attention and subsequent memory storage processes [43]. The P300 has been associated with WM updating [44,45], with a higher P300 amplitude having been associated with lower WM load [37,38,46,47,48,49,50]. The P300 amplitude also increases over the course of WM task performance, which has been deemed to be an effect of training WM [46,51]. This effect may reflect the same phenomenon, since the perceived workload, or cognitive effort required, can be considered to decrease through training [52,53]. An increase in the amplitude of the component has also been observed after both single-session [54,55] and repetitive TDCS [56], suggesting that anodal TDCS could induce similar changes to the P300 amplitude to training and reduced cognitive effort.

Here, we sought to identify potential alterations in electrophysiological processing underlying WM training in combination with anodal TDCS or sham stimulation (Sham). To this end, participants carried out a 2-back task, which elicits well-recognized neural correlates of WM processing.

## 2. Materials and Methods

### 2.1. Participants

Twenty healthy adults aged 21–31 years (M = 26.20, SD = 3.32; 7 females) were recruited via public notice and were compensated with 8 Euro per hour. All participants (N = 20) completed the first two sessions, and 15 participants (TDCS: N = 7; Sham: N = 8) completed all five stimulation sessions. All available data were analyzed (Sessions 1 and 2: N = 20; Session 3: N = 18; Session 4: N = 17; Session 5: N = 15). Sample sizes required to show a difference between conditions varying in cognitive load in an n-back task, reflected in the RTs and P300 amplitudes, were calculated based on values reported in previous studies, at a power of 80% and with a type I error rate of 5%. With group mean RTs of 900 ms and 750 ms and a SD of 100 ms [57], 7 participants would be required. Group means of 595 ms and 708 ms and a SD of 70 ms [37] would also require 7 participants. With respect to P300 amplitudes, group means of 11 μV and 14 μV and a SD of 2 μV [57] indicate a requirement for 7 participants. With group means of 4.2 μV and 5.5 μV and a SD of 0.8 μV [37], 6 participants would be required. We assumed differences between the studies reflect differing paradigm parameters and concluded that our sample size was adequate. All participants were right-handed and had no metallic implants, no history of neurological disorders, including depression, no significant history of drug abuse, no current medication, and no experience with TDCS. All participants gave written informed consent prior to inclusion in the study, which was approved by the Local Ethics Committee of the Medical Faculty of the Otto von Guericke University Magdeburg, in accordance with the Declaration of Helsinki. All participants underwent the KAI Short-Form Intelligence Test [58] to ensure an IQ over 85. The experimental and control groups were matched for age (TDCS: M = 25.8, SD = 3.4; Sham: M = 26.8, SD = 3.4), gender (TDCS: 4 females; Sham: 3 females), and WM capacity (KAI Test; TDCS: M = 119.0, SD = 8.5; Sham: M = 119.6, SD = 11.8) and did not differ significantly in any of these three variables. The study was designed to meet the safety criteria for TDCS in humans [59]. In order to evaluate possible side-effects (SEs), two questionnaires were designed, based on the recommendations of Brunoni et al. [60]. The first was issued directly after each session to address acute symptoms during stimulation (acute SEs), with items covering *tiredness*, *prickling*, *itching*, *headache*, and *nausea*. The second was concerned with longer-lasting symptoms that the participants experienced during the rest of the day and at night (persisting SEs), consisting of the same items and the additional item of *insomnia*.

### 2.2. Study Design and Working Memory Assessment

Each participant was randomly assigned either to the experimental group, to receive TDCS, or to the control group, to receive Sham, and was scheduled to undergo five sessions on five consecutive days (Figure 1c). During each session, the participants performed three runs of a visual 2-back task [61], which were each separated by a five-minute break. Participants were seated in front of a 24-inch computer monitor, with an eye-to-monitor distance of 0.85 m, and presented with one of five white letters (A–E), which were 1.5 cm in size (visual angle 1.0°) and centered on a black screen for 500 ms. During an inter-stimulus period of 1500 ms, a blue fixation cross was presented against the black background screen (Figure 1a). Participants were asked to respond as to whether the current letter was the same as or different to the letter presented two trials before. The task was presented using the software Presentation version 18.2 (Neurobehavioral Systems, Berkeley, CA, USA). Each run took ten minutes and consisted of a series of 300 letters, comprising 75 matching letters (targets) and 225 non-matching letters (non-targets). Participants provided their responses by using their left and right index fingers to press keys assigned as match and non-match (counterbalanced across all participants).

### 2.3. Transcranial Direct Current Stimulation

Anodal direct current was delivered using the battery-driven DC-Stimulator Plus (Serial 2049, version 4.3.00.17, NeuroConn, Ilmenau, Germany) and two 7 × 5 cm rubber electrodes, resulting in a current density of 0.02857 mA/cm^2^. The electrodes were covered with sponges soaked in 0.9% saline solution. The anode was placed over F3, according to the international 10–20 system for EEG, and fixed with an EEG cap (Figure 1b). This location was chosen, because it has been used previously to target the left DLPFC in WM tasks [11,13,62]. The cathode was fixed with a 74 cm × 4 cm rubber strap (NeuroConn, Ilmenau, Germany) to the left shoulder to avoid unwanted cerebral stimulation effects at the cathode [63]. In the TDCS group, the current was ramped up gradually over 15 s at the beginning of the stimulation to prevent electrical transients [1]. Subsequently, stimulation took place for 15 min at 1 mA. During the first five minutes, participants rested, then they performed the 2-back task for the remaining ten minutes. At the end of the stimulation, the current was faded out over an additional 15 s. Stimulation in the Sham group included ramping up of the current for 15 s at the beginning, followed by 30 s of direct current stimulation at 1 mA and ramping down for 15 s for blinding to group allocation [2,64], since tingling side-effects are mostly perceived in the first few seconds of stimulation. The participants were unable to see the stimulation device and its settings and were hence unaware of their group allocation. Group allocation was known to the instructor to enable setting of the stimulator. Care was taken to ensure that the procedure followed was identical for both groups. Similar to the TDCS group, the participants rested for the first five minutes of Sham stimulation.

### 2.4. EEG Recording and Preprocessing

During Sessions 1 and 5, EEG was recorded pre- and post-stimulation during the 2-back task using a 32-electrode EEG cap with electrode locations according to the international 10–20 system. Electrode Fpz was set as the ground, and the data were referenced against the right mastoid. Electrode impedance was kept below 5 kOhm. The sampling rate was 500 Hz. An electrooculogram was recorded with one electrode under the right eye for vertical movement, which was referenced against Fp2, and two electrodes on the canthi of each eye for horizontal movement. The EEG was recorded with Brain Vision Recorder version 1.20.0001 (Brain Products, Gilching, Germany) software, and the following offline data processing was conducted with MATLAB R2016a (MathWorks, Natick, MA, USA) and the toolbox EEGLAB, version 13.6.5b [65].

The EEG data were band-pass filtered from 0.1 Hz to 30 Hz. Following visual inspection, one or more of the channels F7, F8, T7, T8, O9, O10, Oz, and Iz were identified as bad channels in each dataset and therefore excluded from all datasets, leaving the channels Fp1, Fp2, F3, Fz, F4, FC1, FC2, C3, Cz, C4, CP1, CP2, P7, P3, Pz, P4, P8, PO7, PO3, PO4, and PO8 for further processing (Figure 1b). The data were segmented into epochs from 200 ms before until 1990 ms after each letter presentation. The 200 ms before stimulus onset were used for baseline correction. Each letter presentation was considered a stimulus. All epochs lacking a corresponding reaction from the participants were excluded, as were epochs containing large artifacts resulting from movements. Independent component analysis was performed to eliminate eye blinks [65]. ERPs were determined separately for each participant by averaging over trials in which the target was successfully identified (Hit) and additionally over trials in which non-target items were correctly rejected (CR). Additional ERPs with all responses were also created. For visualization, grand average ERPs were then calculated across participants.

### 2.5. Data Analysis

#### 2.5.1. Behavioral Data

To evaluate WM performance, d-prime (d’) was used [66]. The d’ provides a measure of a participant’s ability to discriminate between targets and non-targets, enabling a quantification of WM performance that is relatively independent of age, gender, and intelligence quotient [67]. The d’, mean RTs for Hits, and mean RTs for CRs were calculated separately for each run and participant and normalized to their corresponding baseline values (Session n minus Session 1: pre-stimulation). The changes in performance relative to the pre-stimulation baseline were then compared between the stimulation groups and over time. For statistical analysis, SPSS Statistics 23 (IBM, Armonk, NY, USA) and MATLAB R2016a (MathWorks, Natick, MA, USA) were used.

Three-way mixed ANOVAs were applied to determine within-subject effects with the factors *acute stimulation* (pre- vs. during vs. post-stimulation) for effects within a single session and *repeated stimulation* (Session 1 vs. Session 2 vs. Session 3 vs. Session 4 vs. Session 5) for effects between sessions. The between-subject factor was the *stimulation type* (TDCS vs. Sham). The effects of these factors on RTs and accuracy (d’) were evaluated. Whenever Mauchly’s test indicated that the assumption of sphericity had been violated, the degrees of freedom were corrected using Greenhouse–Geisser estimates of sphericity. Post hoc comparisons were performed using the least significant difference (LSD) test.

#### 2.5.2. Event-Related Potentials

To analyze the effects of repetitive anodal TDCS, Session 5 pre-stimulation recordings were used, as they enabled the examination of long-term effects of repeated stimulation separately from the acute effects of stimulation. ERPs were calculated for a parietal-central region of interest (ROI), consisting of the average ERP waveforms from electrodes Cz, P3, Pz, and P4 (Figure 1b). The peak amplitude of the P300 component was determined by detection of the local maximum relative to the baseline in a pre-defined time window [36]. The time window from 250–500 ms was initially examined [51], but due to individual variance, the window was expanded to 250–570 ms. The time point of the peak was used to determine the P300 latency. For amplitude and latency comparison between the groups (TDCS vs. Sham), the non-parametric Mann–Whitney U test was applied to baseline values (Session 1 pre-stimulation), for acute stimulation (Session 1 post-stimulation), and for repeated stimulation (Session 5 pre-stimulation). Pearson’s correlation test was calculated to evaluate whether there was a relationship between the electrophysiological (P300 amplitude over ERPs derived from all responses) and the behavioral (d’ and RTs over all responses) measures relative to the pre-stimulation baseline (Session 1) separately for the TDCS and Sham groups.

## 3. Results

### 3.1. Reaction Times

Application of a three-way mixed ANOVA for the response type Hit revealed a main effect of *repeated stimulation* (F(1.59, 20.64) = 15.82, *p* < 0.001, ηp^2^ = 0.55). Post hoc comparison showed that the mean RTs were significantly faster as the sessions progressed (i.e., over time) (LSD: *p* < 0.03), except between Sessions 2 and 3 (*p* = 0.18) (Figure 2a). A main effect of *acute stimulation* was also identified (F(1.4, 18.2) = 5.0, *p* = 0.03, ηp^2^ = 0.28). Post hoc comparison indicated that the mean RT post-stimulation (M = −157.33 ms, SD = 34.76 ms) was significantly faster than the mean RT pre-stimulation (M = −128.23 ms, SD = 28.31 ms, LSD: *p* = 0.02) and during stimulation (M = −141.38 ms, SD = 34.76 ms) (LSD: *p* = 0.01). There was no significant effect of *stimulation type* (F(1, 13) = 0.46, *p* = 0.51), and none of the interactions was significant.

The analysis of the response type CR revealed a significant main effect of *repeated stimulation* (F(1.41, 16.92) = 46.66, *p* < 0.001, ηp^2^ = 0.80). Post hoc comparison showed that the mean scores differed significantly between all sessions, becoming faster as the sessions progressed (LSD: *p* < 0.05). A significant main effect of *acute stimulation* (F(1.34, 16.05) = 15.24, *p* = 0.001, ηp^2^ = 0.56) was also found. A post hoc comparison showed differences between all stimulation time points, with mean RTs becoming faster over time. Again, an effect of *stimulation type* (F(1, 12) = 1.42, *p* = 0.26) was not identified, and there were no significant interactions.

### 3.2. Accuracy

Changes in d’ were analyzed analogously to changes in the RTs. A three-way mixed ANOVA showed a significant effect of *repeated stimulation* (F(2.04, 20.43) = 7.05, *p* = 0.005, ηp^2^ = 0.41). Post hoc comparison showed that d’ improved over the sessions (Figure 2b). No main effect of *acute stimulation* was observed (F(2, 20) = 0.23, *p* = 0.80), and there was also no main effect of *stimulation type* (F(1, 10) = 0.12, *p* = 0.73). No significant interactions were detected.

### 3.3. ERPs

The P300 amplitudes for the response type Hit did not differ significantly between the TDCS (Mdn = 8.9) and Sham (Mdn = 7.1) groups before the first stimulation (Mann-Whitney U = 43.0, *p* = 0.73, r = −0.09) (Figure 3a). After repeated stimulation, the P300 amplitude in the TDCS group (Mdn = 11.1) was significantly higher than in the Sham group (Mdn = 8.0) (U = 12.0, *p* = 0.02, r = −0.56) (Figure 3c). After a single stimulation session, however, the P300 amplitudes did not differ significantly between the TDCS (Mdn = 8.2) and Sham groups (Mdn = 9.2; U = 34.0, *p* = 0.72, r = −0.10) (Figure 3b).

The P300 amplitudes for the response type CR did not differ between the TDCS (Mdn = 6.8) and Sham groups (Mdn = 7.7) before the first stimulation (U = 29.0, *p* = 0.16, r = −0.33). Following single-session stimulation, the P300 amplitudes did not differ between the TDCS (Mdn = 8.8) and Sham groups (Mdn = 7.7; U = 37.0, *p* = 0.93, r = −0.03). After repeated stimulation, the P300 amplitude in the TDCS group (Mdn = 6.9) also did not differ from that in the Sham group (Mdn = 5.7) (U = 33.0, *p* = 0.82, r = −0.07). Effects on P300 latency were tested analogously and did not reveal any group differences for the response types Hits or CRs.

### 3.4. Correlation between Working Memory Performance and P300 Amplitude

The change in P300 amplitude in the ROI following single-session stimulation did not correlate with the change in d’ or RTs. However, a significant correlation between d’ and P300 amplitude was observed after repeated stimulation. After four stimulation sessions (Session 5 pre-stimulation), the change in P300 amplitude positively correlated with the change in d’ in the TDCS group (N = 9, r = 0.73, *p* = 0.02) but not in the Sham group (N = 8, r = 0.47, *p* = 0.25). Similarly, a negative correlation was found between the change in RTs and the change in P300 amplitude in the TDCS group (N = 9, r = −0.82, *p* = 0.007) but not in the Sham group (N = 8, r = −0.44, *p* = 0.28).

### 3.5. Side-Effects

Three participants in the TDCS group terminated their participation in the study early due to persisting SEs. One participant developed a burn at the cathode location on the left shoulder after the fourth stimulation session. It presented as a skin lesion with a small blister that had ruptured. The participant had been aware of a burning sensation during the stimulation but had not felt it to be of sufficient severity to warrant reporting and premature termination of the stimulation. Burn marks as a SE of TDCS have been reported previously [68,69]. Another participant had persistent reddening of the scalp the following day after two stimulation sessions. Both participants were advised not to continue, and both lesions resolved completely in the following days. A third participant developed mild dizziness during stimulation, which resolved on each occasion upon terminating the stimulation, and decided against continuation after three stimulation sessions. The remaining participants had no persisting SEs. 

To investigate the frequencies of acute SEs, the symptoms in the questionnaire were grouped together independently of the session numbers. This resulted in a total of 90 sessions, with N = 50 in the TDCS group and N = 40 in the Sham group. Mann–Whitney U tests were performed for each variable, yielding two symptoms occurring significantly more frequently in the TDCS group than in the Sham group, namely *itching* (mean rank: TDCS = 52.28; Sham = 37.03; U = 661.00, *p* = 0.002, r = −0.33) and *prickling* (mean rank: TDCS = 50.82; Sham = 38.85; U = 734.00, *p* = 0.002, r = −0.24).

## 4. Discussion

The effects of repetitive anodal TDCS to the DLPFC on WM performance and its electrophysiological correlates were investigated. The ERP component P300 was evaluated: after four stimulation sessions, a significant increase in P300 amplitude was observed in the TDCS group compared to the Sham group in a parietal ROI. This increase suggests that repetitive anodal TDCS potentially reinforces the electrophysiological mechanisms of training, as an increase in P300 amplitude is associated with training [46,51,56] and decreased WM load [37,38,46,47,48,49,50,70]. Since this effect was not observed in the Sham group, it cannot be attributed to a general training effect, which has been shown in more difficult versions of the n-back task [46,56], nor to a longer training period [46,51,56].

The observed increase in P300 amplitude facilitated by repetitive anodal TDCS to the DLPFC is consistent with enhanced utilization of the DLPFC-PPC network, which is deemed to underpin WM function [71,72,73] and successful WM processing [74]. Correlation between a known electrophysiological correlate of WM processing in the PPC and WM performance provides further support for the notion that the PPC is important for WM performance [20,26,27]. Additional evidence for the involvement of this network has also been provided by transcranial electrical current stimulation studies applying transcranial alternating current stimulation (TACS), which showed improved WM performance following in-phase frontal-parietal theta TACS [75,76,77,78]. The posterior spatial distribution of the P300 is, moreover, in accord with other reports of P300 modulation during WM processing [47,79], in which more posterior distributions were associated with younger participants and with lower WM load. We note that repetitive anodal TDCS had an impact specifically on the P300 amplitude of the response type Hit rather than on successful retrieval in general (Hits and CRs). It has been shown that the P3b is elicited when a rare target stimulus is to be discriminated against a background of frequent standard stimuli [43]. In the current study, the Hit condition only applied to a quarter of the trials and therefore reflects rare target detection.

After single-session stimulation, no significant increase in P300 amplitude was observed. We could therefore not replicate the findings of Keeser et al. [54], who found a significant increase in P300 amplitude following single-session TDCS but not Sham, nor those of studies assessing TDCS effects on the P300 in other tasks [39,80]. It should be noted, however, that all of the aforementioned studies employed longer exposure to a higher anodal current intensity. It has been specifically suggested that ten minutes of 1 mA stimulation applied to the DLPFC might not influence the excitability of the underlying cortex strongly enough to induce measurable effects during the post-stimulation task [81], which could also be the case for 15 min of stimulation with 1 mA. We chose these parameters, however, to minimize the risk of SEs in consecutive stimulation sessions, hypothesizing that an impact on behavioral measures could be achieved with stimulation repetition as opposed to with longer application of a higher current intensity in a single session. The stimulation strength appears to be an important mediator of the effects of TDCS [62,82]. Our findings would suggest that longer exposure to TDCS and/or higher current intensities have a more beneficial outcome compared to exposure to smaller current intensities and/or a shorter exposure time [83]. This possible dose–response curve [83] might explain the mixed behavioral results in the field. The electrophysiological findings in the current study are in accord with the hypothesis of a dose–response curve on the electrophysiological level, as it is plausible that although the protocol applied here was not sufficient to influence the P300 component strongly after a single stimulation, a cumulative effect could be achieved by repetitive anodal TDCS. We postulate that stimulating either once with a higher intensity or repeatedly with a lower intensity could lead to similar results.

Previously, it has also been suggested that improvements in behavioral performance associated with TDCS are more likely in patient groups than in healthy participants [84,85,86]. The reasons for this could be that healthy participant performance is closer to the ceiling level already, but also the baseline levels of cortical excitability, which can affect the outcome of anodal TDCS [87,88]. For example, patients with psychiatric diseases who show symptoms of reduced executive function, such as in major depressive disorder (MDD) and schizophrenia, have shown altered DLPFC activity [89,90,91,92,93], as well as altered functional frontal-parietal connectivity [94]. Anodal TDCS of the DLPFC has shown promising results as a treatment for MDD [17,95,96,97] and schizophrenia [98,99], including improvement of executive function [17,95,99].

We did not find a group difference in accuracy or RTs between the TDCS and Sham groups. However, a significant positive correlation was found in the TDCS group between the change in d’ and the change in P300 amplitude from session 1 to session 5 pre-stimulation and between the change in RTs and the change in P300 amplitude from session 1 to session 5 pre-stimulation but not in the Sham group. These findings are consistent with the observation of a trend towards such a correlation in previous studies investigating the effects of repeated training sessions [51] and single-session anodal TDCS [55], which suggests a possible relationship between stimulation repetition, the electrophysiological reaction, and the observable behavioral changes.

Further studies using TDCS and investigating electrophysiological changes in EEG are required, considering the lack of clear behavioral changes, despite the correlation between electrophysiological parameters and behavior found in the current study. In particular, the possible dose–response curve should be explored on an electrophysiological level in a larger cohort, varying the stimulation intensity and also the time interval between stimulation sessions. Usage of repetitive stimulation sessions within a single day could also be considered, based on a study in which spatial WM response speed was particularly enhanced when repeated anodal PPC TDCS was performed within the time period in which stimulation-related performance enhancement still persisted [100]. We note, however, that the potential SEs related to repetitive stimulation could be increased with a more intensive stimulation program. While a recent systematic review suggested that repeated TDCS sessions were not associated with significantly greater risk to participants, the authors did suggest that under-reporting could not be ruled out [55]. In the current study, in which the participants were blinded with a common stimulation protocol [64], our TDCS group did experience significantly more acute SE compared to the control group. This finding adds to a recently started discussion about the effectiveness of current blinding methods [101,102].

Limitations of the current study include the use of a single cognitive load and stimulation intensity [31,55]. Also, the current findings should be regarded with caution, based on the number of participants, given the potentially small effect size of TDCS on WM, generally requiring a larger sample size for identifying behavioral changes [103]. Under-powered studies can lead to an increase in false negatives and false positives [103,104]. Despite the power calculations suggesting that we have an adequate number of participants to address our hypotheses, the number of participants is nonetheless small, and our findings should be regarded as preliminary, needing a larger sample size to replicate the findings.

## 5. Conclusions

The aim of the current study was to investigate the influence of single-session and repetitive anodal TDCS on WM and explore the underlying mechanisms of action. Although the behavioral measures provided negligible support for an enhancing effect of TDCS on WM performance, the ERP findings, and moreover their correlation with behavioral performance, suggest effects of repetitive anodal TDCS on relevant electrophysiological processes underlying WM processing. Repetitive anodal TDCS of the DLPFC influenced the frontal–parietal network, involving both the DLPFC and the PPC, which provides additional evidence that this network is involved in WM processing.

## Figures and Tables

**Figure 1 brainsci-12-01545-f001:**
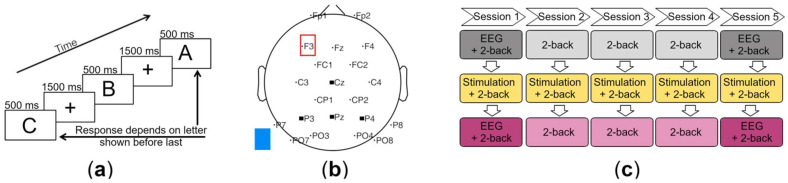
(**a**) The 2-back task. (**b**) EEG channel locations. Black squares mark the electrodes of the parietal-central region of interest (Cz, P3, Pz, and P4). The red square marks the anode over F3 as the stimulation electrode, and the blue square marks the cathode as the extracephalic reference electrode. (**c**) Course of the experiment over five consecutive days.

**Figure 2 brainsci-12-01545-f002:**
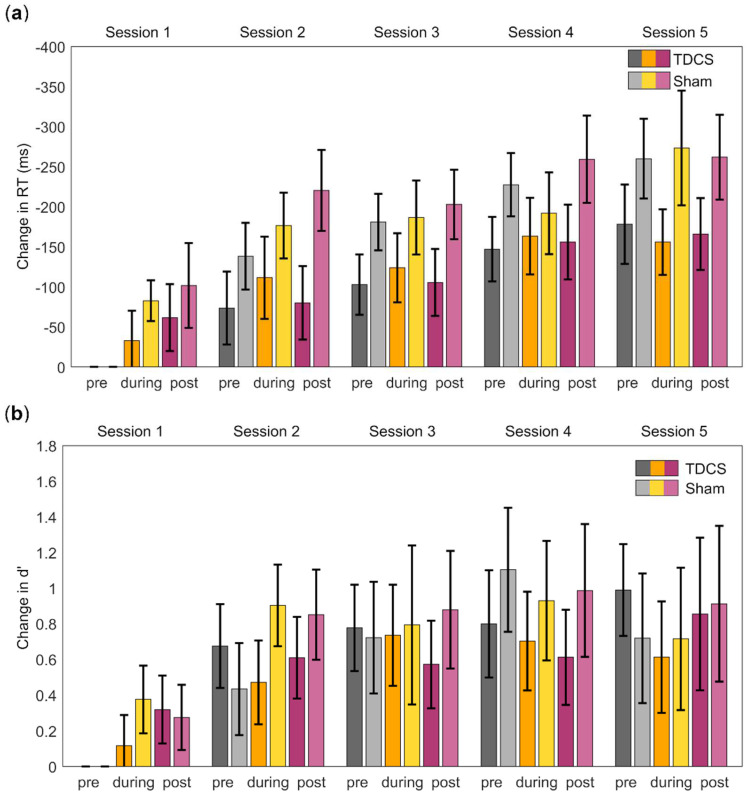
Performance during the 2-back task, measured by reaction times (RTs) and d-prime (d’), before, during, and after stimulation over the five sessions. (**a**) Change in RTs in ms for the response type Hit relative to Session 1 pre-stimulation. (**b**) Change in d’ relative to Session 1 pre-stimulation. Error bars = standard error of the mean. Both measures indicated improved performance over time but no significant difference between the groups.

**Figure 3 brainsci-12-01545-f003:**
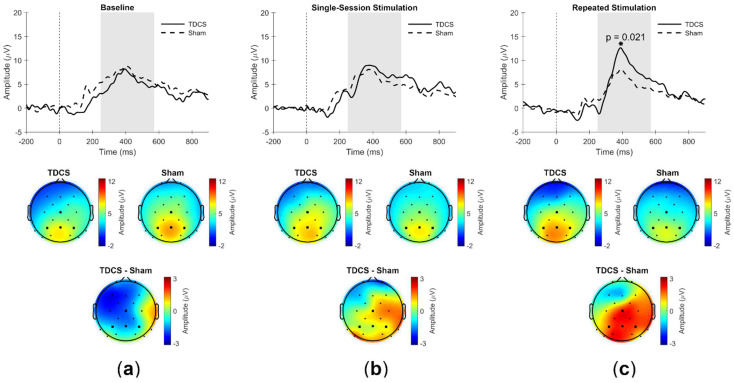
Grand average event-related potentials (ERPs) of the region of interest (Cz, P3, Pz, and P4) for TDCS and Sham groups during Hits (top), including topographical plots averaged over the time window 250–590 ms, based on the grand average ERPs across participants, and their difference (TDCS minus Sham, bottom) (**a**) at the pre-stimulation baseline, (**b**) after single-session stimulation, and (**c**) after repeated stimulation. Shading = P300 time window (250–590 ms), during which the local maximum identified was deemed to reflect the P300 component. * = Statistically significant difference between TDCS and Sham groups (*p* = 0.021).

## Data Availability

The data that support the findings of this study are available from the corresponding author upon reasonable request.

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
