# Peer review of "Repetitive Anodal TDCS to the Frontal Cortex Increases the P300 during Working Memory Processing"

_brainsci, 2022, doi:10.3390/brainsci12111545_

Round 1
Reviewer 1 Report
The proposed manuscript addresses the important topic of the neurophysiological effects of tDCS. It could provide a contribution to understanding the mechanisms of action of tDCS in terms of changes in neurophysiology and thus in the cognitive and behavioral dimensions.
Minor changes may be useful:
1. Please attach a sample size calculation and/or outline the expected power of the trial with this sample size.
2. Is there a blinding of subject and investigator? Please describe blinding procedure more in detail.
3. Abstract: Line 16: Effects could be „potentiated“ is a little too much in my opinion – better “effects are expected to be increased”
4. Introduction: Line 33: tDCS does not cause direct depolarization of the neurons but a change in the resting membrane potential towards depolarization. Please make this more clear.
5. Methods: Eligibility criteria: What about history of psychiatric diseases besides drug abuse like depression as we know that in patients with major depressive disorder the baseline DLPFC activity is suggested to be altered compared to healthy individuals. Did participants take any medication?
6. Discussion: Please discuss the small sample size and the differences between healthy subjects and patients with psychiatric disorders
Reviewer 2 Report
This study aims to investigate the effects of transcranial direct current stimulation (TDCS) on working memory (WM). One major finding is, the behavioral performances were significantly influenced by neither single session nor multi-session TDCS, while another major finding is, the amplitude of P300 signals increased after repetitive TDCS, which implies relevant neural circuit was influenced during the process. The limitation of the current study mostly comes from the fact that the effect size of TDCS on WM is relatively small. Overall, I think the manuscript is well written and the findings contribute to the understanding of the effects of TDCS on brain activity. Below are my comments.
In section 3.4 it is showed that the change in P300 amplitude positively correlated with the change in d’ for the TDCS group. What about reaction time (RT)? Does the change of RT correlate with the change of P300 amplitude?
Consider to change the color code in Figure 1 (c). The order of the three experimental blocks for each session (day) is clear so doesn’t need to be color coded. It would be better to use colors to represent the types of experimental blocks. For example, you can keep all “Stimulation + 2-back” blocks yellow, and use grey for all “2-back” blocks on different days, and red for all “EEG + 2-back” sessions.
In conclusion you mentioned that the current results provide additional evidence that the frontal-parietal network is involved in WM processing. Please add a few citations about the previous studies showing the relationship between this network and WM.
